# Evaluation of Murrell’s EKF-Based Attitude Estimation Algorithm for Exploiting Multiple Attitude Sensor Configurations

**DOI:** 10.3390/s21196450

**Published:** 2021-09-27

**Authors:** Sharanabasaweshwara Asundi, Norman Fitz-Coy, Haniph Latchman

**Affiliations:** 1Department of Mechanical and Aerospace Engineering, Old Dominion University, Norfolk, VA 23529, USA; 2Department of Mechanical and Aerospace Engineering, University of Florida, Gainesville, FL 32611, USA; nfc@ufl.edu; 3Faculty of Engineering, The University of the West Indies, Mona, Kingston 7, Jamaica; haniph.latchman@uwimona.edu.jm

**Keywords:** picosatellites, nanosatelllites, extended Kalman filter, EKF, Murrell’s EKF, satellite attitude estimation

## Abstract

Pico- and nano-satellites, due to their form factor and size, are limited in accommodating multiple or redundant attitude sensors. For such satellites, Murrell’s implementation of the extended Kalman filter (EKF) can be exploited to accommodate multiple sensor configurations from a set of non redundant attitude sensors. The paper describes such an implementation involving a sun sensor suite and a magnetometer as attitude sensors. The implementation exploits Murrell’s EKF to enable three sensor configurations, which can be operationally commanded, for satellite attitude estimation. Among the three attitude estimation schemes, (i) sun sensor suite and magnetometer, (ii) magnetic field vector and its time derivative and (iii) magnetic field vector, it is shown that the third configuration is better suited for attitude estimation in terms of precision and accuracy, but can consume more time to converge than the other two.

## 1. Introduction

Pico- and nanosatellites are being increasingly considered for space missions due to their reduced cost and development time. These class of satellites, conceived as educational tools and experimental platforms, have complemented their traditional counterparts and proved success as utility platforms [1,2,3,4]. However, due to the constrained size, weight, power generation, and storage capability, the design and development of these class of satellites has been challenging. These constraints have forced the designer to revisit the design process and explore some of the techniques and tools used in the earliest satellites [5]. To increase their potential utility, pico- and nanosatellites may be required to host capabilities such as precision attitude determination and control, among others. A space mission relying on such a capability for success, would traditionally accommodate redundancies. However, for a pico- or a nano-satellite platform, accommodating redundancies may not be an option. In such a scenario, innovative implementation and operation to accommodate redundancies from a set of non-redundant systems, is higly desirable. For a remote sensing mission, such as a wild-life monitoring project, the use of pico- or nanosatellite platforms as against traditional satellites, may be economically more viable and technically adequate. For such a mission, a redundant attitude determination system from a set of non-redundant attitude sensors, would be highly desirable. This paper revisits the efficient implementation and utility of Murrell’s version of the extended Kalman filter (EKF), to enable three sensor configurations from a set of non redundant attitude sensors.

The design and development of the attitude determination system (ADS) and attitude estimation with focus on the rational for adopting Murrell’s implementation of EKF, were considered for SwampSat [6,7,8,9,10,11], a CubeSat [12,13,14] class pico-satellite. It was a technology demonstration space mission to validate on orbit rapid retargeting and precision pointing capability for pico-/nano-satellites with an experimental ACS consisting of four single gimbal control moment gyroscopes (CMGs) [8,15,16]. SwampSat mission was required to be designed and developed as per the CubeSat design specification/standard [14], which imposed significant size, weight, and power (SWaP) constraints. The on-orbit power generating capability (∼1.5W) limited the continuous operation of its high performance processor selected for precision computing. The flight computer, although suited well for routine operations, was limited in its ability to support the computational requirements of attitude determination and control. To address these limitations, the computing platform was designed as a distributed system. An MSP430 based flight computer is in continuous operation and the CMG controller, a high performance digital signal processor from Texas Instruments, operated intermittently to perform attitude operations. The distributed computing architecture and its operational design in the form of command and data handling system and telemetry formulation, were adapted for SwampSat, whose power requirements for proving the mission were significantly larger than the on-orbit average power generated.

The paper describes the distribution of ADS hardware across two computing platforms, which are adopted to address the power limitation of pico-satellites. The description also details the sensor suite selected for this mission and the models adopted for inclusion in the attitude estimation implementation. Furthermore, the specific implementation and its utility to accommodate multiple configurations are described. Finally, the simulations carried out to evaluate these configurations, their performance results and the comparison of these configurations are presented.

## 2. Attitude Determination Hardware and Sensor Models

Pico- and nano-satellites, which need to host capabilities such as precision attitude determination and control, may require precision sensors, actuators, complex algorithms, and high speed processors. It can be challenging to accommodate such enhanced capabilities on a pico-satellite with limited power generation capability. The pico-satellite addressed in this paper is one such satellite and a distributed computing architecture, shown in Figure 1, is adopted to efficiently utilize the limited power resources and effectively operate all of the on board capabilities. The schematic, shown in Figure 1, identifies the main hardware and software components of the attitude determination and estimation system, which are distributed across the two computing platforms. The pico-satellite is designed to accommodate modified light-to-voltage converters (LTV) as coarse sun sensors (CSSs) and a low power magnetometer, which are both interfaced to the flight computer. Each CSS on the pico-satellite outputs the average direction of a light source, which is the sun in this case. The magnetometer is designed to output the sum of the ambient magnetic field of interest, which is the Earth’s magnetic field in this case. A micro electromechanical system (MEMS) based inertial measurement unit (IMU) is interfaced to the high speed auxiliary computer (CMG controller). The rate gyros in the IMU output the angular rates of the pico-satellite in an inertial frame [17]. A detailed design of this distributed computing architecture is described in Reference [11].

### 2.1. Attitude Sensors

Although LTVs as coarse sun sensors and MEMS based magnetometers are limited in their ability to precisely determine the reference vectors in the satellite body frame, they are conveniently available in sizes suitable for pico-satellite applications and their impact on the satellite power budget is significantly low. If the error sources of these sensors can be mathematically modeled and incorporated, they can be effectively used for attitude vector computations. A sensor model, based on the design and analysis of an LTV based sun sensor [18], is considered for evaluating the attitude estimation design. The magnetometer, an HMC2003 unit from Honeywell [19], was selected as the other attitude sensor due to its suitability for pico-satellites and its effectiveness in low Earth orbits. The maximum magnetic field strength experienced on the Earth’s surface is limited to 0.6 Gauss, as calculated by the International Geophysical Data Center, and the HMC2003 is capable of measuring up to ±2 gauss at a resolution of 40μgauss. The latest versions of this series of magnetometers from Honeywell are the HMC2003 magnetic hybrid and the HMR2300 smart digital magnetometer [20]. The mathematical model considered for these attitude sensors is discussed below.

Let ${}^{B}R_{i}$ represent the *i*-th measurement of a vector in frame $F_{B}$ and ${}^{I}R_{i}$ its representation in a reference frame $F_{I}$. The discrete time measurement of the vector and its representation in both frames can be modeled as follows [21]
(1)$$\begin{array}{c} {}^{B}R_{i}={\underset{=}{C}}_{BI}{}^{I}R_{i}+\vartheta_{i} \end{array}$$
where ${\underset{=}{C}}_{BI}$ is the attitude matrix, which transforms the vectors in inertial frame to satellite body frame. The sensor error vector $\vartheta_{i}$ is Gaussian with the following properties.
(2)$$\begin{array}{ccc} E\{\vartheta_{i}\} & = & 0 \end{array}$$
(3)$$\begin{array}{ccc} E\{\vartheta_{i}\vartheta_{i}^{T}\} & = & \sigma_{i}^{2}I \end{array}$$
where E{} denotes the expectation.

#### 2.1.1. Sun Sensor

The voltage measured from the Sun sensor [7,18] on the *i*-th face of the pico-satellite is modeled using a cosine profile as,
(4)$$\begin{array}{c} {\tilde{V}}_{i}=\left\{\begin{array}{c} \begin{array}{cc} {}^{B}{\hat{R}}_{i}.^{B}{\hat{n}}_{i}+\eta_{s}\left(1{-}^{B}{\hat{R}}_{i}.^{B}{\hat{n}}_{i}\right) & \;{\mathrm{if}}^{B}{\hat{R}}_{i}.^{B}{\hat{n}}_{i}\geq 0\\ 0 & \;\mathrm{otherwise} \end{array} \end{array}\right. \end{array}$$
where $\eta_{s}$ is a zero-mean normally distributed random number with standard deviation $\sigma_{s}=0.1$ V. It was observed that the voltage generated matched the cosine profile more closely when the angle of incidence was closer to zero [18]. Accordingly, the sun vector in the body frame is expressed as,
(5)$${}^{B}R_{i}=\frac{1}{{\left(\sum_{i=1}^{6}{{\tilde{V}}^{2}}_{i}\right)}^{0.5}}\sum_{i=1}^{6}{\tilde{V}}_{i}{}^{B}{\hat{n}}_{i}$$

#### 2.1.2. Magnetometer

The magnetic field vector is modeled according to the following expression,
(6)$${}^{B}R_{i}={\underset{=}{C}}_{BI}{}^{I}R_{i}+\eta_{m}+\beta_{m}$$
where $\eta_{m}$ is a white random vector with the standard deviation of each component $\sigma_{m}=0.005{∥}^{B}R_{i}∥$ and $\beta_{m}$ is a bias vector incorporating the influence of spacecraft environment.

#### 2.1.3. Inertial Measurement Unit and Gyro Model

The ADIS16405 [22], a MEMS based IMU from Analog Devices, was identified for measuring the body rates of the pico-satellite. The latest version of this series of IMU from Analog Devices is the ADIS16495 [23]. The IMU is equipped with gyros operating in rate mode and can be sampled at frequencies of up to 350 Hz. The ADIS16405 was selected based on a trade study of MEMS based inertial sensors [8]. The low power constraint, port available to interface with the onboard computer, operating temperature range and the size/mass were some of the characteristics evaluated for the selection process. The ADIS16405 sensor has an operating temperature range of −40 ${}^{\circ }$C to +85 ${}^{\circ }$C, interfaces with the on board computer through the serial peripheral interface (SPI) port and consumes 33 mA @ 5 V when operating in a normal mode. The unit weighs 16 g and is ∼23×23×23 mm in dimension. For evaluating the attitude estimation design, a mathematical model of the gyros developed by Farrenkopf [24] and applied by Hoffman and McElroy [25] is considered. The gyro model relating the gyro output vector $\tilde{\omega }$, the satellite angular velocity ω, drift rate bias β and drift rate noise $\eta_{v}$ is expressed as
(7)$$\begin{array}{ccc} \tilde{\omega } & = & \omega +\beta +\eta_{v} \end{array}$$
(8)$$\begin{array}{ccc} \dot{\beta } & = & \eta_{u} \end{array}$$
where $\eta_{v}$ and $\eta_{u}$ are uncorrelated zero-mean Gaussian white-noise processes with standard deviations of $\sigma_{v}=0.007$ deg/s ${}^{2}$, $\sigma_{u}=2$ deg/h, respectively, and having the same properties as the attitude sensor error vector $\vartheta_{i}$. The values of standard deviations are obtained from the sensor data sheet [22]. The IMU was subjected to experimentation on a rotary table with precision encoders to verify the sensor integrity and determine the initial bias characteristics. A setup of the precision rotary table and the plots verifying the integrity of the gyros are shown in Figure 2. To determine the initial bias characteristics, the IMU was placed on a stationary platform and the gyro readings were sampled for a period of 60 min. The mean of the sampled gyro measurements about each axes were adopted as the initial gyro bias.

## 3. Attitude Estimation and Sensor Configurations

The attitude estimation algorithm, shown in Figure 3, is designed as a 6-state EKF to be implemented in two phases—(i) attitude determination and (ii) attitude propagation. The two phase implementation is adopted to maximize the frequency of the satellite attitude initialization through the flight computer. While the auxiliary processor is waiting to hear from the flight computer, it iterates through the attitude propagation phase. A handshaking protocol checks for the availability of attitude sensor measurements from the flight computer to re-initialize the satellite attitude. The attitude estimation states, which include three attitude errors and three gyro drift rate biases, are propagated as a 6 × 6 error covariance matrix. The three components of attitude error incorporate the inaccuracies in the attitude sensors and the gyro drift rate bias in the MEMS gyros. The attitude estimation algorithm is evaluated for three different attitude sensor configurations: (i) Attitude estimation using sun sensors and magnetometer, (ii) attitude estimation using the magnetic field vector and its time derivative, and (iii) attitude estimation using the magnetic field vector. All three configurations employ gyros for measuring the satellite angular rates. Murrell’s version of the EKF is adopted to accommodate the three sensor configurations and operationally command a specific configuration from ground, if required. The evaluations of the three configurations are discussed below.

### 3.1. Attitude Estimation Using Sun Sensors and Magnetometer

During the attitude determination phase, the error covariance matrix, $P_{k}^{-}$, is initialized with values determined from experiments, models and component data sheets of the attitude sensors and gyros. The initial bias of the gyros, ${\hat{\beta }}_{k}^{-}$, is assigned with values determined from experiments. The sun and magnetic field vector are acquired in the body and inertial frames to initialize the satellite attitude quaternion, ${\hat{\bar{q}}}_{k}^{-}$, from QUEST [26]. The corresponding attitude matrix, ${\underset{=}{C}}_{BI}$ and the vector part of the quaternion are computed and stored as separate variables for use in the algorithm. Since two measurements are used to compute the attitude quaternion, the algorithm performs two iterations of Equations (Equation 9)–(Equation 13) to calculate the sensitivity matrix $H_{k}$ (Equation (Equation 9)), Kalman gain $K_{k}$ (Equation (Equation 10)), error covariance $P_{k}^{+}$ (Equation (Equation 11)), residual $\epsilon_{k}$ (Equation (Equation 12)) and error state $\Delta {\hat{\tilde{x}}}_{k}^{+}$ (Equation (Equation 13)). The parameters computed in Equations (Equation 9)–(Equation 13) are used to update the quaternion ${\hat{\bar{q}}}_{k}^{+}$ (Equation (Equation 15)), gyro bias ${\hat{\beta }}_{k}^{+}$ (Equation (Equation 16)) and error covariance matrix ${\hat{P}}_{k}^{+}$.
(9)$$\begin{array}{ccc} H_{k} & = & \left[\begin{array}{cc} {\left[{\underset{=}{C}}_{BI}{}^{I}R_{i}\right]}^{\times } & 0_{3\times 3} \end{array}\right] \end{array}$$
(10)$$\begin{array}{ccc} K_{k} & = & P_{k}^{-}H_{k}^{T}{\left[H_{k}P_{k}^{-}H_{k}^{T}+\sigma_{i}^{2}I\right]}^{-1} \end{array}$$
(11)$$\begin{array}{ccc} P_{k}^{+} & = & \left[I-K_{k}H_{k}\right]P_{k}^{-} \end{array}$$
(12)$$\begin{array}{ccc} \epsilon_{k} & = & {(}^{B}{\tilde{R}}_{i}-{\underset{=}{C}}_{BI}{}^{I}R_{i}){|}_{t_{k}} \end{array}$$
(13)$$\begin{array}{ccc} \Delta {\hat{\tilde{x}}}_{k}^{+} & = & \Delta {\hat{\tilde{x}}}_{k}^{-}+K_{k}\left[\epsilon_{k}-H_{k}\Delta {\hat{\tilde{x}}}_{k}^{-}\right] \end{array}$$
(14)$$\begin{array}{ccc} \Delta {\hat{\tilde{x}}}_{k}^{+} & = & {\left[\delta {\hat{\alpha }}_{k}^{+T}\Delta {\hat{\beta }}_{k}^{+T}\right]}^{T} \end{array}$$
(15)$$\begin{array}{ccc} {\hat{\bar{q}}}_{k}^{+} & = & {\hat{\bar{q}}}_{k}^{-}+\frac{1}{2}\Xi \left({\hat{\bar{q}}}_{k}^{-}\right)\delta {\hat{\alpha }}_{k}^{+} \end{array}$$
(16)$$\begin{array}{ccc} {\hat{\beta }}_{k}^{+} & = & {\hat{\beta }}_{k}^{-}+\Delta {\hat{\beta }}_{k}^{+} \end{array}$$

The quaternion, gyro bias and the error covariance matrix computed during the attitude determination phase are passed as arguments to the attitude propagation phase. The satellite angular rates are acquired from the gyro measurements and the gyro bias is compensated in Equation (Equation 17). The estimated satellite angular rates are used within this phase to propagate the satellite attitude using Equation (Equation 18).
(17)$$\begin{array}{ccc} {\hat{\omega }}_{k}^{+} & = & {\tilde{\omega }}_{k}-{\hat{\beta }}_{k}^{+} \end{array}$$
(18)$$\begin{array}{ccc} {\hat{\bar{q}}}_{k+1}^{-} & = & \bar{\Omega }\left({\hat{\omega }}_{k}^{+}\right){\hat{\bar{q}}}_{k}^{+} \end{array}$$
where,
(19)$$\begin{array}{ccc} \bar{\Omega }\left({\hat{\omega }}_{k}^{+}\right) & = & \left[\begin{array}{cc} cos(\frac{1}{2}\left|\right|{\hat{\omega }}_{k}^{+}\left||\Delta t\right)I_{3\times 3}-{\left[{\hat{\psi }}_{k}^{+}\right]}^{\times } & {\hat{\psi }}_{k}^{+}\\ -{\hat{\psi }}_{k}^{+T} & cos(\frac{1}{2}||{\hat{\omega }}_{k}^{+}||\Delta t) \end{array}\right] \end{array}$$
(20)$$\begin{array}{ccc} {\hat{\psi }}_{k}^{+} & = & \frac{sin(\frac{1}{2}\left|\right|{\hat{\omega }}_{k}^{+}\left||\Delta t\right){\hat{\omega }}_{k}^{+}}{\left|\right|{\hat{\omega }}_{k}^{+}\left|\right|} \end{array}$$

Additionally, the error covariance matrix is also propagated using Equation (Equation 9) through Equation (Equation 13) during this phase to update the states of the satellite.
(21)$$\begin{array}{c} P_{k+1}^{-}=\Phi_{k}P_{k}^{+}\Phi_{k}^{T}+G_{k}Q_{k}G_{K}^{T} \end{array}$$
where,
(22)$$\begin{array}{ccc} G_{k} & = & \left[\begin{array}{cc} -I_{3\times 3} & 0_{3\times 3}\\ 0_{3\times 3} & I_{3\times 3} \end{array}\right] \end{array}$$
(23)$$\begin{array}{ccc} Q_{k} & = & \left[\begin{array}{cc} \left(\sigma_{v}^{2}\Delta t+\frac{1}{3}\sigma_{u}^{2}\Delta t^{3}\right)I_{3\times 3} & -\left(\frac{1}{2}\sigma_{u}^{2}\Delta t^{2}\right)I_{3\times 3}\\ -\left(\frac{1}{2}\sigma_{u}^{2}\Delta t^{2}\right)I_{3\times 3} & \left(\sigma_{u}^{2}\Delta t\right)I_{3\times 3} \end{array}\right] \end{array}$$
(24)$$\begin{array}{ccc} \Phi & = & \left[\begin{array}{cc} \Phi_{11} & \Phi_{12}\\ \Phi_{21} & \Phi_{22} \end{array}\right] \end{array}$$
(25)$$\begin{array}{ccc} \Phi_{11} & = & I_{3\times 3}-{\left[\hat{\omega }\right]}^{\times }\frac{sin(||\hat{\omega })||\Delta t)}{\left|\right|\hat{\omega }\left|\right|}+{\left[\hat{\omega }\right]}^{\times^{2}}\frac{\left\{1-cos(|\right|\hat{\omega }\left||\Delta t)\right\}}{\left|\right|\hat{\omega }{\left|\right|}^{2}} \end{array}$$
(26)$$\begin{array}{ccc} \Phi_{12} & = & {\left[\hat{\omega }\right]}^{\times }\frac{\left\{1-cos(|\right|\hat{\omega }\left||\Delta t)\right\}}{\left|\right|\hat{\omega }{\left|\right|}^{2}}-I_{3\times 3}\Delta t-{\left[\hat{\omega }\right]}^{\times }\frac{\left\{|\right|\hat{\omega }||\Delta t-sin(||\hat{\omega }\left||\Delta t\right)}{\left|\right|\hat{\omega }{\left|\right|}^{3}} \end{array}$$
(27)$$\begin{array}{ccc} \Phi_{21} & = & 0_{3\times 3} \end{array}$$
(28)$$\begin{array}{ccc} \Phi_{22} & = & I_{3\times 3} \end{array}$$

Here, $\sigma_{v}^{2}$ and $\sigma_{u}^{2}$ are the variances associated with the random drift $\eta_{v}$ and drift rate ramp $\eta_{u}$ components of the on-board gyros. The components are discussed as part of the gyro model. The sampling interval is captured by the parameter Δt. The filter is implemented in two phases to account for the distributed implementation of the attitude determination and estimation system. During the attitude determination phase, measurements are obtained from the attitude sensors, which are interfaced to the flight computer. During the attitude propagation phase, measurements are obtained from the on-board gyros which are interfaced to the CMG controller. The propagation phase when operated independently from the determination phase can propagate the initialized attitude until a new set of attitude measurements are available.

### 3.2. Attitude Estimation Using Magnetic Field Vector and Its Time Derivative

To mitigate any limitations in the on-board sun sensors, attitude estimation using the magnetic field vector and its first time derivative is explored. The estimation scheme is designed to accommodate this particular sensor configuration. If ${\underset{=}{C}}_{BI}$ represents the transformation from inertial reference frame to satellite body frame, i.e.,
(29)$$\begin{array}{c} {}^{B}\left(\right)={\underset{=}{C}}_{BI}{}^{I}\left(\right) \end{array}$$
ω is the angular velocity of the satellite body frame with respect to the inertial frame, i.e.,
(30)$$\begin{array}{c} {\overset{˙}{\underset{=}{C}}}_{BI}=-{\left[\omega \right]}^{\times }{\underset{=}{C}}_{BI} \end{array}$$

Let B be the magnetic field vector measured in both frames, then
(31)$$\begin{array}{c} {}^{I}B\left(t\right)={\underset{=}{C}}_{BI}^{T}\left(t\right){}^{B}B\left(t\right) \end{array}$$

If B is measured at times $t_{1}$ and $t_{2}$, then
(32)$$\begin{array}{ccc} {}^{I}B\left(t_{2}\right) & = & {\underset{=}{C}}_{BI}^{T}\left(t_{2}\right){}^{B}B\left(t_{2}\right) \end{array}$$
(33)$$\begin{array}{ccc} {}^{I}B\left(t_{1}\right) & = & {\underset{=}{C}}_{BI}^{T}\left(t_{1}\right){}^{B}B\left(t_{1}\right) \end{array}$$

Furthermore,
(34)$$\begin{array}{ccc} {\underset{=}{C}}_{BI}\left(t_{2}\right) & = & {\underset{=}{\Delta C}}_{BI}\left(t_{2}-t_{1}\right){\underset{=}{C}}_{BI}\left(t_{1}\right) \end{array}$$
(35)$$\begin{array}{ccc} {\underset{=}{C}}_{BI}\left(t_{2}\right) & = & \left(1-{\left[\Delta \theta \left(t_{1}\right)\right]}^{\times }\right){\underset{=}{C}}_{BI}\left(t_{1}\right) \end{array}$$

Conversely, it can be stated (i.e., reversing time propagation)
(36)$$\begin{array}{ccc} {\underset{=}{C}}_{BI}\left(t_{1}\right) & = & {\underset{=}{\Delta C}}_{BI}\left(t_{1}-t_{2}\right){\underset{=}{C}}_{BI}\left(t_{2}\right) \end{array}$$
(37)$$\begin{array}{ccc} {\underset{=}{C}}_{BI}\left(t_{1}\right) & = & \left(1-{\left[\Delta \theta \left(t_{2}\right)\right]}^{\times }\right){\underset{=}{C}}_{BI}\left(t_{2}\right) \end{array}$$
where,
(38)$$\begin{array}{ccc} \Delta \theta (t_{2}) & = & -\Delta \theta (t_{1}) \end{array}$$
(39)$$\begin{array}{ccc} {\underset{=}{C}}_{BI}\left(t_{1}\right) & = & \left(1+{\left[\Delta \theta \left(t_{1}\right)\right]}^{\times }\right){\underset{=}{C}}_{BI}\left(t_{2}\right) \end{array}$$

Thus,
(40)$$\begin{array}{ccc} {}^{I}B\left(t_{2}\right) & = & {\underset{=}{C}}_{BI}^{T}\left(t_{2}\right){}^{B}B\left(t_{2}\right) \end{array}$$
(41)$$\begin{array}{ccc} {}^{I}B\left(t_{1}\right) & = & {\underset{=}{C}}_{BI}^{T}\left(t_{2}\right)\left(1-{\left[\Delta \theta \left(t_{1}\right)\right]}^{\times }\right){}^{B}B\left(t_{1}\right) \end{array}$$
(42)$$\begin{array}{ccc} {}^{I}B\left(t_{2}\right){-}^{I}B\left(t_{1}\right) & = & {\underset{=}{C}}_{BI}^{T}\left(t_{2}\right){[}^{B}B\left(t_{2}\right){-}^{B}B\left(t_{1}\right)+{\left[\Delta \theta \left(t_{1}\right)\right]}^{\times }{}^{B}B\left(t_{1}\right)] \end{array}$$

Dividing by $\Delta t=t_{2}-t_{1}$,
(43)$$\begin{array}{ccc} \frac{{}^{I}B\left(t_{2}\right){-}^{I}B\left(t_{1}\right)}{\Delta t} & = & {\underset{=}{C}}_{BI}^{T}\left(t_{2}\right)\left[\frac{{}^{B}B\left(t_{2}\right){-}^{B}B\left(t_{1}\right)}{\Delta t}+\frac{\Delta \theta (t_{1})}{\Delta t}{}^{B}B\left(t_{1}\right)\right] \end{array}$$
(44)$$\begin{array}{ccc} \frac{{}^{I}B\left(t_{2}\right){-}^{I}B\left(t_{1}\right)}{\Delta t} & = & {\underset{=}{C}}_{BI}^{T}\left(t_{2}\right)\left[\frac{{}^{B}B\left(t_{2}\right){-}^{B}B\left(t_{1}\right)}{\Delta t}+{\left[\omega \right]}^{\times }\left(t_{1}\right){}^{B}B\left(t_{1}\right)\right] \end{array}$$

Pre-multiplying Equation (Equation 31) and the derivative form of Equation (44) by the transformation matrix ${\underset{=}{C}}_{BI}$, where $\dot{B}=\frac{B\left(t_{2}\right)-B\left(t_{1}\right)}{\Delta t}$, it can be concluded that the magnetic field vector B and its first time derivative $\dot{B}$ have the following relationship [27,28].
(45)$$\begin{array}{ccc} {\underset{=}{C}}_{BI}{}^{I}B & = & {}^{B}B \end{array}$$
(46)$$\begin{array}{ccc} {\underset{=}{C}}_{BI}{}^{I}\dot{B} & = & {}^{B}\dot{B}+{\left[\omega \right]}^{\times }{}^{B}B \end{array}$$

This relationship can be exploited to acquire two vectors for attitude determination. Within the attitude estimation algorithm, the magnetic field and its first time derivative are acquired in the inertial and body frame and passed as arguments to QUEST for initializing the satellite attitude quaternion. The satellite angular velocity with respect to the inertial frame coordinatized in the body axes is acquired from the on-board gyros for computing the complete time derivative of the magnetic field vector. The rest of the estimation algorithm is executed as described in the flowchart shown in Figure 3.

### 3.3. Attitude Estimation Using Magnetic Field Vector

Although the deterministic approach described for the above configuration is, in theory, better suited for attitude determination and propagation the non-linearities associated with the magnetic field vector can significantly influence the time derivative vector and subsequently the attitude estimate. To address the limitations of this approach, attitude estimation using a magnetic field as the only attitude vector is explored. The use of Murrell’s version of the EKF has proved to be advantageous and an efficient approach for accommodating this sensor configuration. QUEST being an optimal estimator requires at least two vectors to compute an attitude estimate. In its place, Shuster’s method of constructing a suboptimal attitude quaternion from a single vector in two frames is adopted for initializing the satellite attitude [29]. The expression for constructing the suboptimal attitude quaternion is
(47)$$\begin{array}{c} \bar{q}=\sqrt{\frac{1{+}^{B}B.^{I}B}{2}}\left[\begin{array}{c} \left(\frac{{}^{B}B\times^{I}B}{1{+}^{B}B.^{I}B}\right)\\ 1 \end{array}\right] \end{array}$$

The initialized attitude quaternion is iteratively processed through Equation (Equation 9) to Equation (Equation 21) for estimating the satellite attitude and gyro bias. Unlike the above two configurations, Equation (Equation 9) through Equation (Equation 13) are updated only once. The remainder of the estimation algorithm is executed as in the above two configurations. The simulations carried out to evaluate the performance of each of these sensor configurations, and their results are described in the following section.

## 4. Simulation and Results

To evaluate the effectiveness of the attitude estimation algorithms described above, simulations are performed with data generated from AGI’s Satellite Tool Kit (STK) [30]. An STK scenario emulating a pico-satellite in a low Earth orbit with the parameters shown in Table 1, having angular rates of up to ±3 deg/s about each axis, is constructed. The data generated for this STK scenario consists of the quantities shown in Table 2. The true quaternion in the data set is used for comparing the estimated quaternion from the attitude estimation algorithm. The attitude data set for each scenario is generated for 100 min with a 25 Hz sampling frequency. The simulation environment is modeled similarly to that discussed in Reference [7]. To emulate sensor measurements on orbit, the data is corrupted with bias and noise parameters. Based on the design and analysis of the sun sensors used for the SwampSat mission [18], the sun vector measurements are corrupted with Gaussian white random noise with zero mean and a standard deviation of $\sigma_{s}=0.1$ V. The magnetic field measurements are corrupted with similar random noise components with zero mean and a standard deviation of $\sigma_{m}=0.005{∥}^{B}R_{i}∥$. The gyro measurements are modeled as described previously with standard deviations of $\sigma_{v}=0.007$ deg/s ${}^{2}$, $\sigma_{u}=2$ deg/h. As simulation parameters, the standard deviation of attitude error covariance is set to 3.4 deg, obtained from sensor data sheet and the standard deviation of gyro drift covariance is set to [0.03110.03030.0272] deg/s, which is experimentally determined. The initial bias for each axis, determined experimentally, is set to [−0.0724−0.19270.0205] deg/s.

To baseline the attitude estimation results for the three scenarios described above, a deterministic attitude determination approach is utilized to compute the satellite attitude from the corrupted sensor measurement. The satellite attitude quaternion, determined from this approach is compared against the true quaternion obtained from STK. The results of this comparison are presented as plots in Figure 4. As part of the results, the effect of varying the sampling frequencies of the attitude sensor measurements is also captured. The impact of a distributed computing implementation is captured as four subplots by varying the sampling frequencies of the attitude sensor measurements. It can be seen from the plots shown in Figure 4 that the attitude error has an approximate upper bound of 21 deg when sampled at 12 Hz and 24 deg when sampled at 1 Hz. It can observed from these plots that the coarse attitude sensors are significantly limited in their ability to determine the satellite attitude, precisely.

### 4.1. Estimation Results Using Sun Sensors and Magnetometer

The results of the attitude estimation simulations performed using sun sensors and magnetometer are shown in Figure 5, Figure 6 and Figure 7. The error in attitude accuracy is captured as an eigen angle and eigen axis of the error quaternion, the error being computed between the estimated quaternion and the true quaternion as described in Reference [31]. The error in angular rates is determined by taking the finite difference between the components of the estimated angular rates and true angular rates. With the available onboard attitude sensors, the estimation algorithm improves upon the attitude determination results shown in Figure 4. The simulation parameters used for this scenario are summarized in Table 3. It can be seen from Figure 5 and Figure 6 that the accuracy of the attitude estimate and estimate angular rates increase with increase in the sampling frequency of the attitude sensors. The distributed implementation of the attitude determination and estimation system enables the gyros to be sampled at a higher rate than the attitude sensors. The subplots shown in Figure 5 investigate the effect of varying sampling frequency of the attitude sensors, while maintaining the sampling frequency of the gyros at 25 Hz. It can be observed that approximately 1 deg pointing accuracy can be achieved if the attitude sensors can be sampled at 12 Hz with the gyros being sampled at 25 Hz. The simulations are carried out for 100 min to qualify and evaluate the estimation algorithm. The estimation algorithm requires a finite time to converge to a steady error, which can be accommodated in the system. The plots shown in Figure 6 and Figure 7 capture the error in estimated angular rates about each axis of the satellite and the error in each of the estimated quaternion components. The error in each of these components can be observed to decrease with an increase in the sampling frequency of the attitude sensors.

### 4.2. Estimation Results Using Magnetic Field Vector

To address any possible limitations in the Sun sensor and enable attitude estimation during eclipse time, simulations are carried out to investigate the performance of the EKF using magnetic field vectors from the magnetometer and gyros. The simulation parameters used for this scenario are summarized in Table 4. The results of the simulations are shown in Figure 8, Figure 9 and Figure 10. The performance is evaluated in terms of attitude accuracy, error in estimated satellite angular rates and error in estimated quaternion components. It can be seen by comparing the plots in Figure 5 and Figure 8 that the EKF in this sensor configuration takes a longer time to converge to a steady state. The attitude estimate, though, has better accuracies compared to those in the previous configuration, particularly with decrease in the sampling rate of the attitude sensors. Since a quaternion and its negative represent the same attitude [32], the convergence of its scalar component $q_{4}=\cos\theta /2$ to a value of 1 or −1 implies the same result.

### 4.3. Estimation Results Using Magnetic Field Vector and Its Time Derivative

Attitude estimation using the magnetic field vector and its first time derivative was first proposed by Natanson et al. in 1991 [28]. It was observed that the Earth’s magnetic field was constantly changing and could be used as a vector for computing the attitude of satellites, particularly those in low Earth orbits. To evaluate the performance of this configuration, simulations are performed, and the results are compared to the above two configurations. The simulation parameters used for this scenario are summarized in Table 5. The results of the attitude estimation are shown in Figure 11, Figure 12 and Figure 13. It can be seen from the plots that the EKF outperforms for the previous two configurations compared to this one. The poor performance of the EKF for this particular configuration may be attributed to the influence of nonlinearity of the magnetic field vector on its time derivative. Additionally, the attitude determination part of the algorithm uses spacecraft angular velocity acquired from the onboard gyros to compute its derivative vector. Although filtered gyro measurements are used to compute the derivate vector, the residual noise and bias influence vector calculation and the a priori attitude estimate. The attitude accuracy achieved using the magnetic field vector and its time derivate approach is on the same order as that achieved using the other two methods.

## 5. Conclusions

An algorithm adapted for estimating the attitude of a pico-satellite and enabling multiple sensor configurations from a set of non-redundant attitude sensors is discussed. Murrell’s version of the EKF is selected to improve upon the computational efficiency [31,33] and accommodate multiple sensor configurations. The QUEST algorithm and the suboptimal estimator are used for initializing the attitude estimate, thereby enabling a faster convergence with the true attitude. The simulation results shown in Figure 4, Figure 5, Figure 8 and Figure 11 justify the need and effectiveness of attitude estimation for a pico-satellite hosting the type of sensors discussed above. The EKF performance for the three sensor configurations is presented in Table 6.

As presented in Table 6, the magnetometer-based attitude estimation is better suited than the other two approaches for a pico-satellite hosting the type of attitude sensors discussed above. Except for the convergence time, the approach is shown to have a relatively better performance in terms of computational resources, power consumption and overall attitude accuracy. As the sampling frequency of the attitude sensors is reduced, the degradation in attitude accuracy is less for magnetometer-based attitude estimation than the other two configurations. An added advantage of using the magnetometer based attitude estimation is that it can be performed during eclipse time. A level of redundancy can be built by exploiting the use of different sensor configurations. To account for a possible failure or uncertainties associated with a sensor type, the attitude estimation algorithm, implemented in Murrell’s form, can be operationally commanded from ground to use the best suited sensor configuration.

## 6. Future Work

For pico-satellites designed with magnetometer as a primary attitude sensor, its onboard calibration is of paramount importance. If the satellite’s attitude knowledge is primarily a function of the magnetometer measurements, the attitude accuracy itself cannot be used to compensate for biases, scale factors or other alignment discrepancies. Coupled with a continuously changing satellite magnetic field of relatively high magnitude, real time calibration and bias compensation can be challenging. Although much work has been published for real-time [34,35] and non real-time [36,37] onboard calibration of magnetometers, there is a need to conduct research for mitigating the adverse effects of spacecraft magnetic field of the same order in magnitude as the ambient field to be measured. For a compactly designed pico-satellite, hosting high power DC motors as the limbs of a CMG pyramid, the utility of a realtime state estimation method, such as that described in Reference [34], is limited and may not be able to completely address the calibration deficiency. The batch methods adopted for offline or non real time calibration are not applicable for use in a satellite with a continuously changing magnetic field. In such a case, a more effective pre-estimation technique to compensate for the satellite magnetic field may be to explore the dual-magnetometer based method.

## Figures and Tables

**Figure 1 sensors-21-06450-f001:**
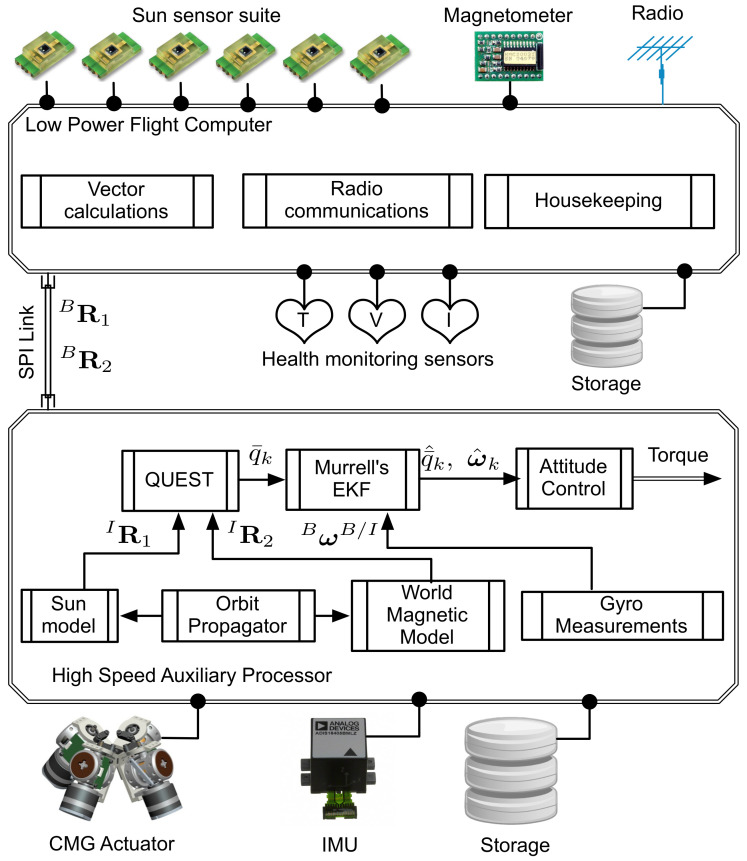
Attitude determination hardware layout.Reprinted with permission from ref. [11] Copyright 2013 Copyright IEEE.

**Figure 2 sensors-21-06450-f002:**
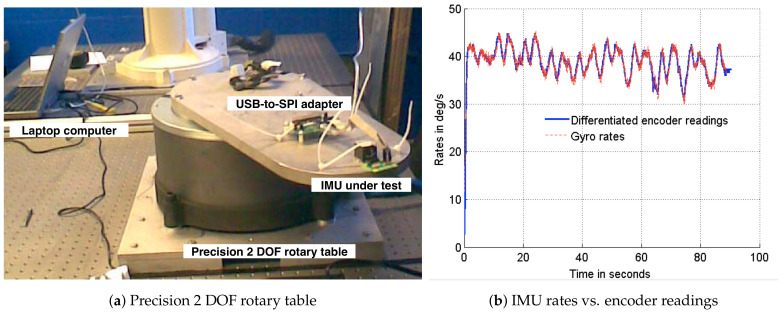
IMU calibration experiment.

**Figure 3 sensors-21-06450-f003:**
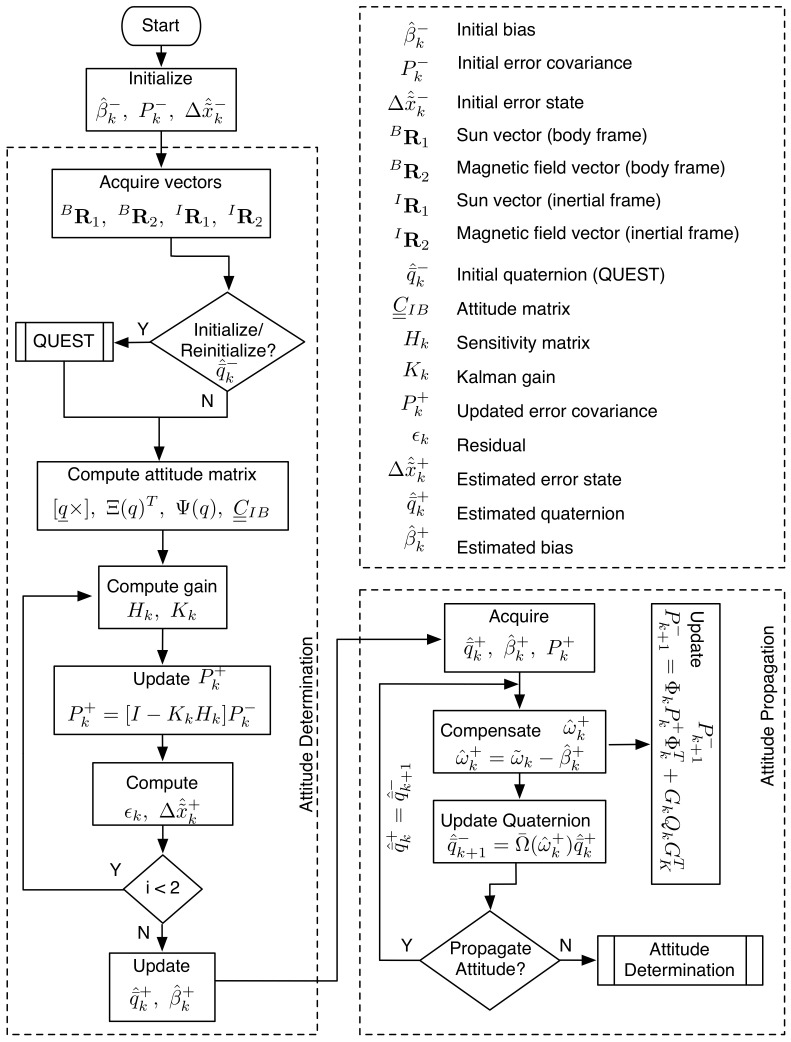
Attitude estimation algorithm for SwampSat.

**Figure 4 sensors-21-06450-f004:**
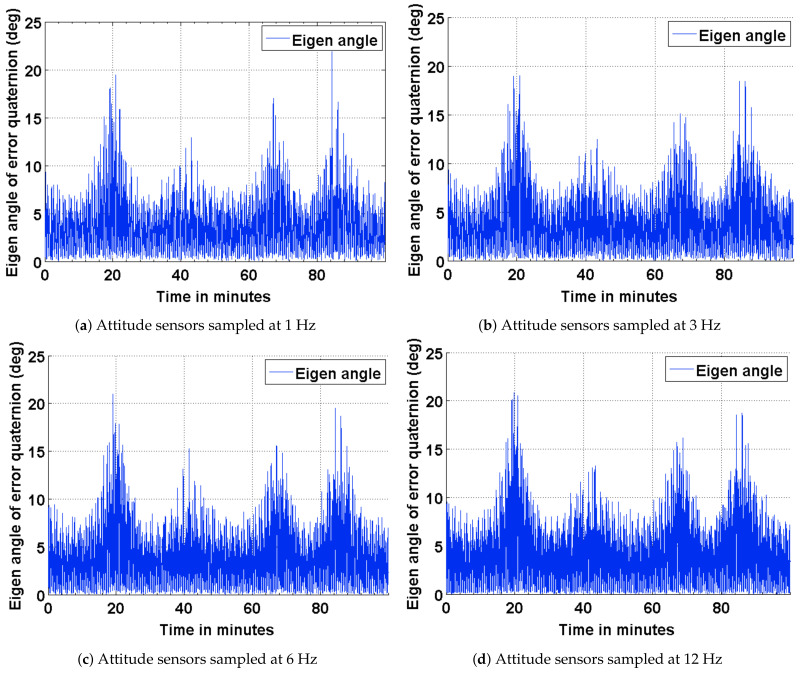
ADS error in terms of the eigen angle of error quaternion.

**Figure 5 sensors-21-06450-f005:**
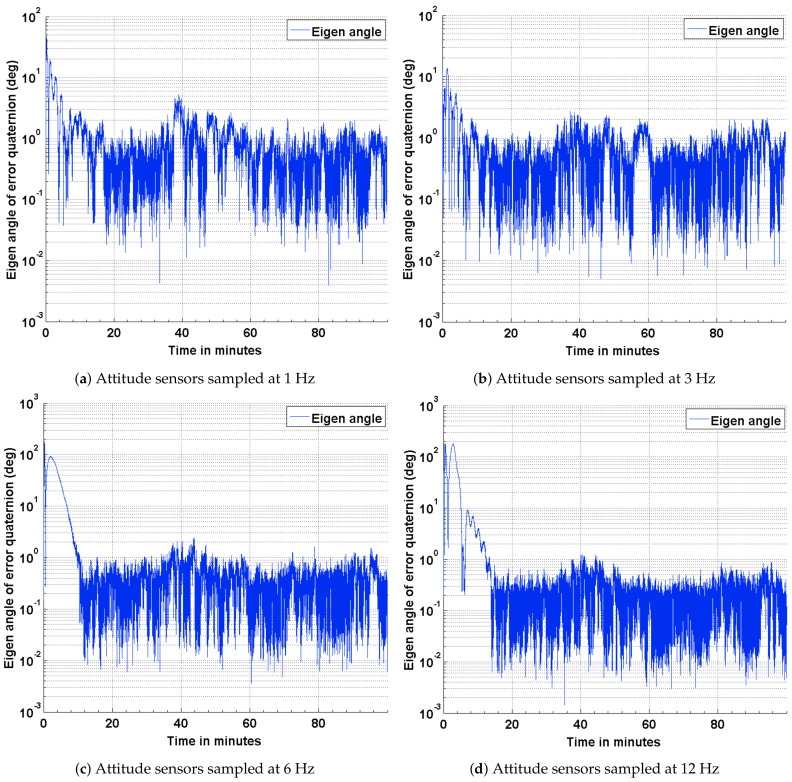
Estimation error in terms of the eigen angle of error quaternion.

**Figure 6 sensors-21-06450-f006:**
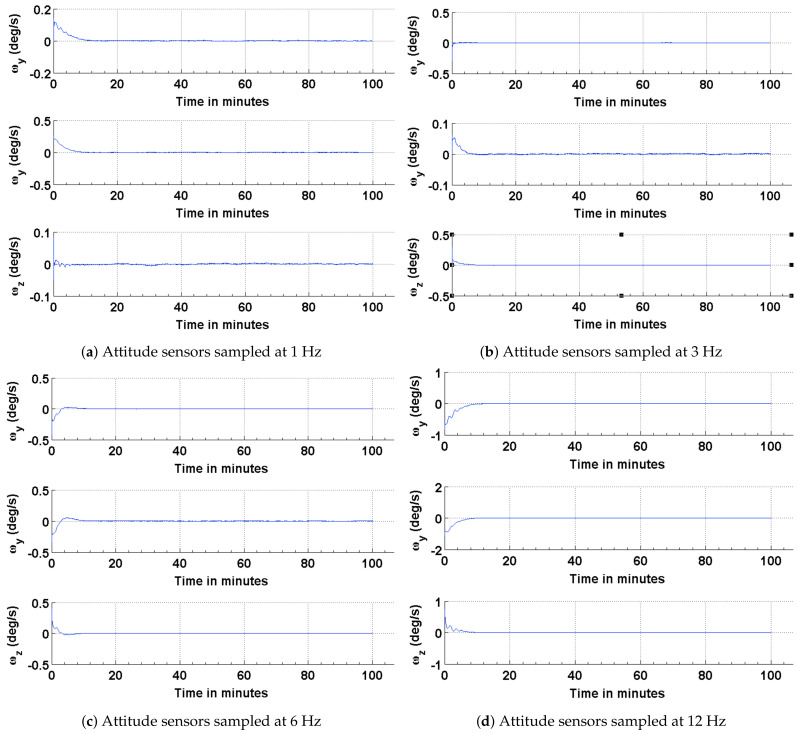
Error in the estimated and true angular rates.

**Figure 7 sensors-21-06450-f007:**
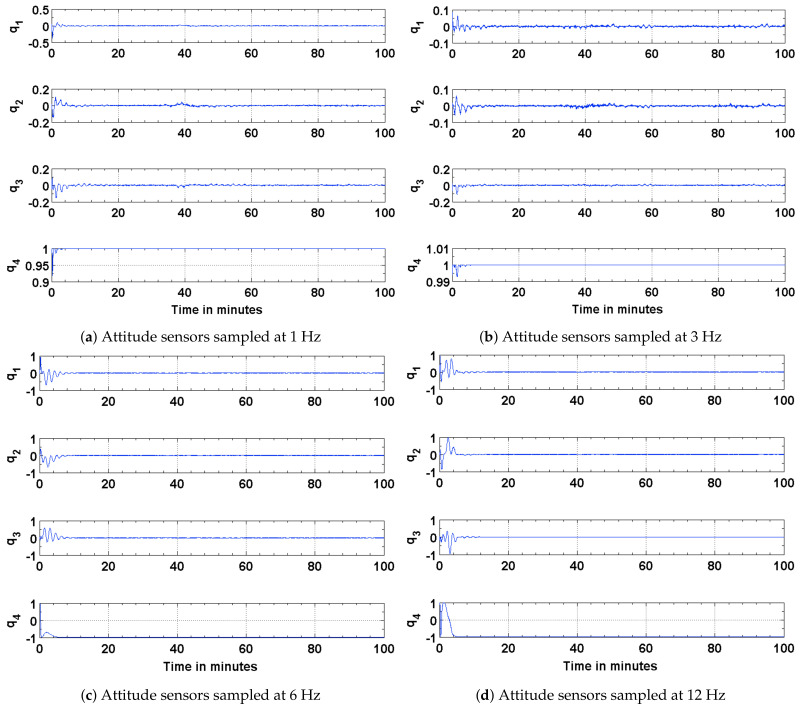
Error in the estimated and true quaternion components.

**Figure 8 sensors-21-06450-f008:**
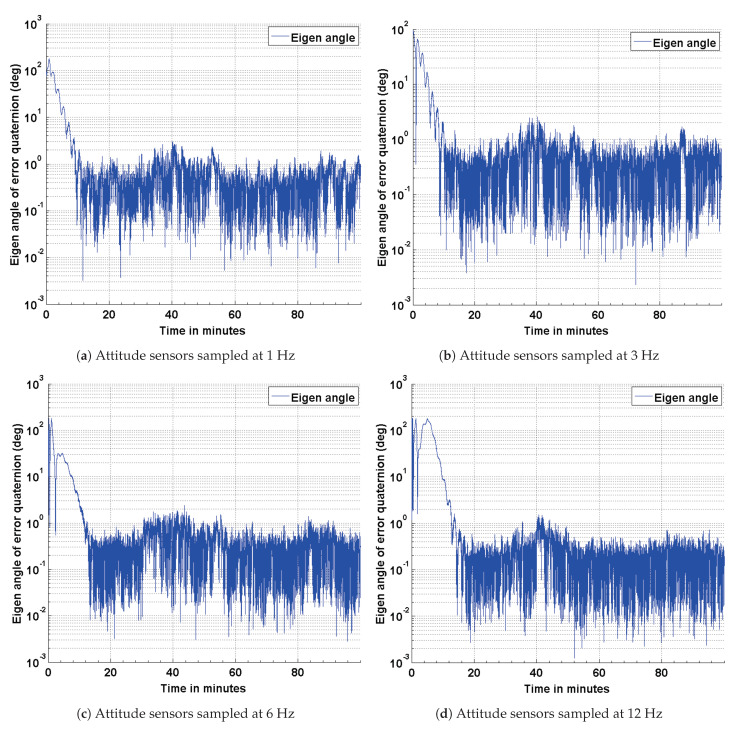
Estimation error in terms of the eigen angle of error quaternion.

**Figure 9 sensors-21-06450-f009:**
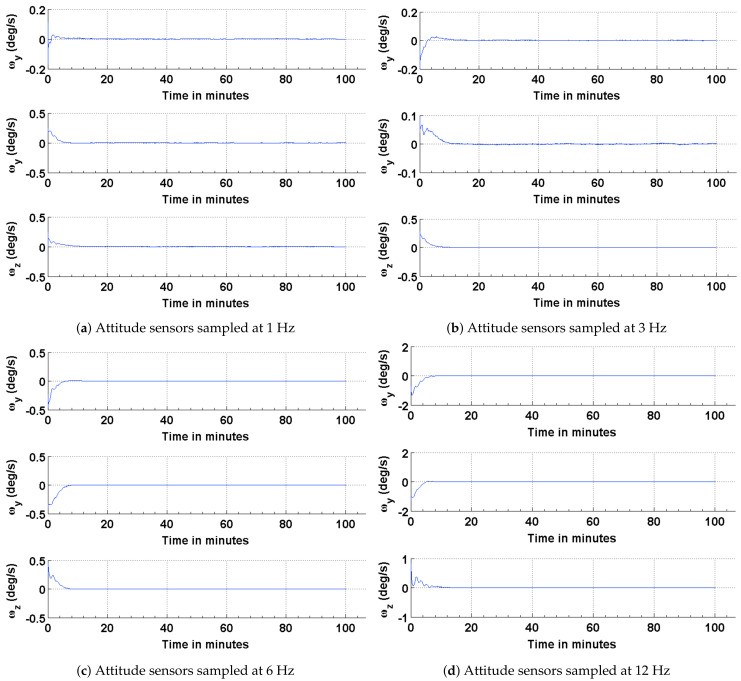
Error in the estimated and true angular rates.

**Figure 10 sensors-21-06450-f010:**
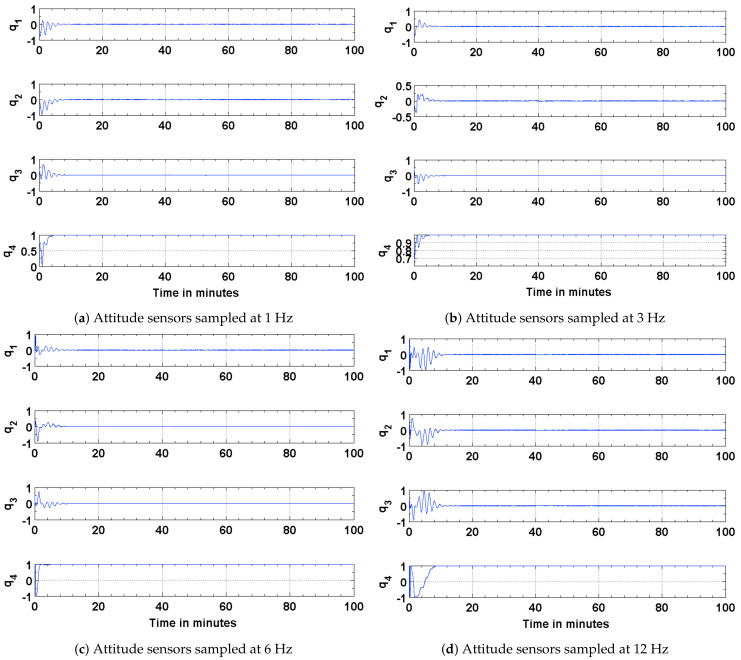
Error in the estimated and true quaternion components.

**Figure 11 sensors-21-06450-f011:**
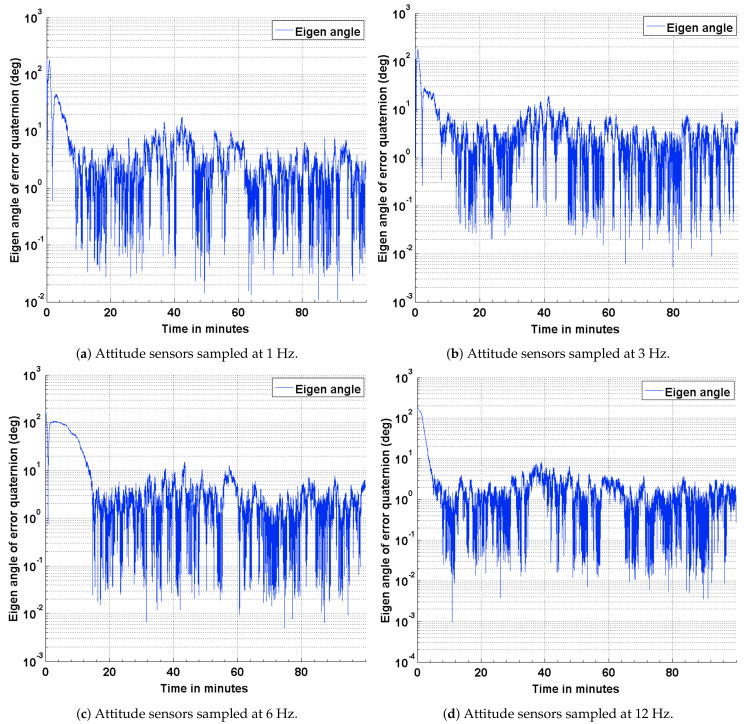
Estimation error in terms of the eigen angle of error quaternion.

**Figure 12 sensors-21-06450-f012:**
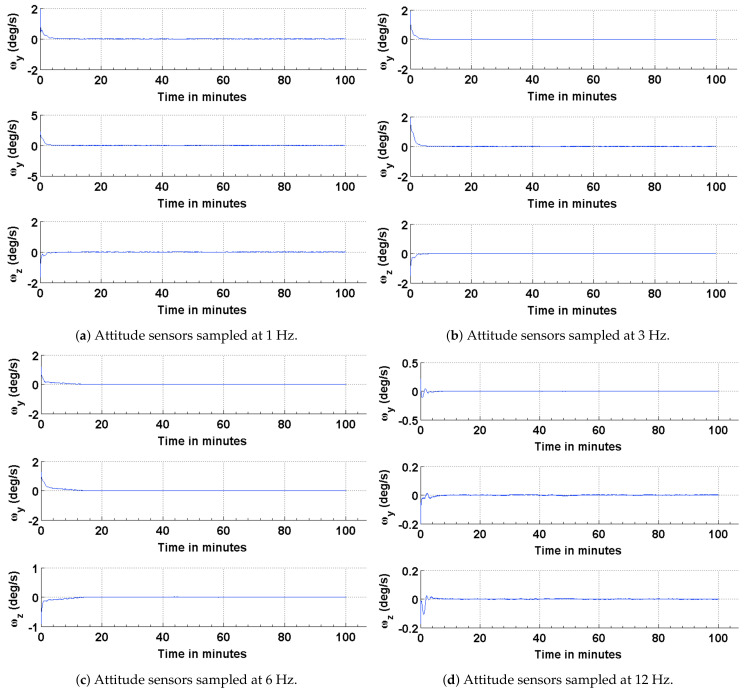
Error in the estimated and true angular rates.

**Figure 13 sensors-21-06450-f013:**
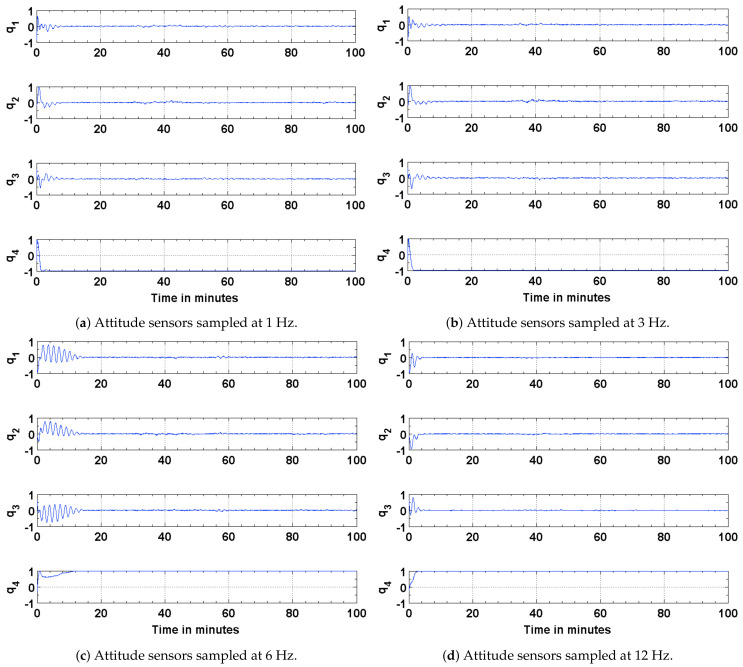
Error in the estimated and true quaternion components.

**Table 1 sensors-21-06450-t001:** LEO parameters for STK scenario.

Parameter	Value
Semi-major axis (a)	6703.14 Km
Eccentricity (e)	0.00
Inclination (i)	51.60 deg
Argument of perigee (ω)	0.00 deg
Longitude of ascending node (Ω)	0.00 deg
True anomaly (ϑ)	0.00 deg

**Table 2 sensors-21-06450-t002:** Attitude data generated for simulation.

Quantity	Description
${}^{B}R_{1}$	3 × 1 magnetic field vector coordinatized in body frame
${}^{B}R_{2}$	3 × 1 Sun vector coordinatized in body frame
${}^{I}R_{1}$	3 × 1 magnetic field vector coordinatized in ECI frame
${}^{I}R_{2}$	3 × 1 Sun vector coordinatized in ECI frame
${}^{B}\omega$	3 × 1 angular velocity vector coordinatized in body frame
$\bar{q}$	4 × 1 quaternion representing true attitude

**Table 3 sensors-21-06450-t003:** Simulation parameters for attitude estimation using sun sensors and magnetometer.

Parameter/Variable	Value
$P_{k}^{-}$	$\left[0.03142^{2},0.03142^{2},0.03142^{2},0.0311^{2},0.0303^{2},0.0272^{2}\right]$
${\hat{\beta }}_{k}^{-}$	${\left[-0.0724,-0.1927,0.0205\right]}^{T}$
$\left[\sigma_{1}\sigma_{2}\right]$	[100,5]
$\left[\sigma_{U}\sigma_{V}\right]$	[0.001×π/180,1×π/180/3600]

**Table 4 sensors-21-06450-t004:** Simulation parameters for attitude estimation using magnetic field vector.

Parameter/Variable	Value
$P_{k}^{-}$	$\left[0.03142^{2},0.03142^{2},0.03142^{2},0.0311^{2},0.0303^{2},0.0272^{2}\right]$
${\hat{\beta }}_{k}^{-}$	${\left[-0.0724,-0.1927,0.0205\right]}^{T}$
σ	175
$\left[\sigma_{U}\sigma_{V}\right]$	[0.001×π/180,1×π/180/3600]

**Table 5 sensors-21-06450-t005:** Simulation parameters for attitude estimation using magnetic field vector and its derivative.

Parameter/Variable	Value
$P_{k}^{-}$	$\left[0.003491^{2},0.003491^{2},0.003491^{2},0.0311^{2},0.0303^{2},0.0272^{2}\right]$
${\hat{\beta }}_{k}^{-}$	${\left[-0.85,-1.1205,0.7205\right]}^{T}$
$\left[\sigma_{1},\sigma_{2}\right]$	[200,2000]
$\left[\sigma_{U}\sigma_{V}\right]$	[0.001×π/180,1×π/180/3600]

**Table 6 sensors-21-06450-t006:** EKF performance comparison for three sensor configurations.

	Magnetometer & Sun Sensors	1 Magnetic Field Vector	2 Magnetic Field Vectors
Attitude accuracy (@ 12 Hz)	<1.5 deg	<1.5 deg	<8 deg
Attitude accuracy (@ 1 Hz)	<5 deg	<3 deg	∼ 20 deg
Converging Time (minutes)	∼5	∼10	∼8
Computational iterations	Two	One	Two
Equations (Equation 9)–(Equation 13)			
Eclipse time estimation	Not possible	Possible	Possible
Sensor power consumption (mW)	360	285	300

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
