# Peer review of "Evaluation of Murrell’s EKF-Based Attitude Estimation Algorithm for Exploiting Multiple Attitude Sensor Configurations"

_sensors, 2021, doi:10.3390/s21196450_

Round 1

Reviewer 1 Report

This paper introduces a method for small satellite attitude control using Murrell's EKF.  The method is most useful for pico and nano-satellites where a comprehensive attitude control system would be too expensive and a low cost system is desirable.  The paper is publishable after the following comments are addressed:

1) In the introduction/background, it would be helpful to explain what the Sun Sensor, Magnetometer, and other components do for the attitude determination and control.  The paper went in details on the math but an overview what the functions are for each of the components would be helpful for the readers, especially for those who are not familiar with the topic.

2) The conclusion section should be shorter.  As a general rule, the conclusion section should be based on results presented in the paper.  Discussion of results and figures should be moved to the results section and should not be in the conclusion section. No new materials should be introduced in the conclusions.

Author Response

Dear Reviewer,

Thank you for taking the time to review our work and provide your valuable feedback. The authors have addressed your review comments to the best of their ability and request you to accept the submission.

Please see the attachment for a point-by-point response.

Sincerely,

Authors.

Reviewer 2 Report

The paper is a comparison of three onboard attitude determination methods for a picosatellite, tested against a  simulated attitude history produced by a commercial software package. The work looks like an elaboration of an activity performed long ago for some specific spacecraft (which is however not presented); e.g., the authors state that the MEMS IMU was selected on the basis of a study done in 2008, and this sensor turns out to be indeed out of production, so that there is little connection to the current technology state of the art.

The scope of the work is quite narrow and the scientific content is of very limited novelty. The conclusions drawn by the authors are hardly of a general nature, being drawn on the basis of essentially a single numeric simulation with little or no support of real data.

I suggest to consider the following points:

Line 49: Missing reference number

Line 59: Instead of a vendor’s website, please reference the manufacturer’s (as correctly done for Ref. 25): https://aerospace.honeywell.com/us/en/learn/products/sensors/3-axis-magnetometer

Equations and formulas should be numbered starting from the very first one on pag. 3

Lines 67 and 68: Define all the symbols used in the formulas

Line 70: “The Sun sensors on each face of the pico-satellite are modeled individually” - The configuration of the spacecarft and its mission should be introduced first; what shape is the satellite? What is it for? How large? The name of the satellite (SwampSat) is only first mentioned in the caption of Fig. 3 ...

Line 93: Figure number is missing

Lne 107: “This form of implementation is adopted to maximize the satellite attitude initialization from the attitude sensors hosted on the flight computer” - Please clarify what is to be maximized.

Line 167: “.... until a new set of set of attitude measurements  ...” - ‘set of’ is repeated twice (typo)

Line 213: What attitude disturbances are included in the STK simulation? Given the low orbital altitude,  some information about, e.g., the shape and optical properties of the satellite bus and the atmosphere model should be provided for the reader to understand what torque disturbances are being considered.

Line 221: “... a standard deviation σs = 0.1 V” - Is this voltage output from the sun sensors? Please clarify; it would also be appropriate to spexificy what is the full range value, i.e. the maximum expected value of this quantity.

Line 224: For angle measurements in degrees, I suggest to use “deg” instead of “ ° ” throughout the paper.

Line 235: “capture” - should be “captured”.

Author Response

(The authors gave the same response as above.)

Round 2

Reviewer 2 Report

The authors corrected some of the deficiencies of the first version. I suggest checking carefully the spelling of the entire text.